# Infection after Anterior Cruciate Ligament Reconstruction: A Narrative Review of the Literature

**DOI:** 10.3390/healthcare12090894

**Published:** 2024-04-25

**Authors:** Giuseppe Danilo Cassano, Lorenzo Moretti, Giovanni Vicenti, Claudio Buono, Federica Albano, Teresa Ladogana, Igor Rausa, Angela Notarnicola, Giuseppe Solarino

**Affiliations:** Orthopaedic & Trauma Unit, Department of Traslational Biomedicine and Neuroscience (DiBraiN), School of Medicine, University of Bari Aldo Moro, AOU Consorziale “Policlinico”, 70124 Bari, Italy; giuseppe.cassano@unifg.it (G.D.C.); lorenzo.moretti@libero.it (L.M.); giovanni.vicenti@uniba.it (G.V.); f.albano34@studenti.uniba.it (F.A.); t.ladogana@studenti.uniba.it (T.L.); i.rausa@studenti.uniba.it (I.R.); angela.notarnicola@uniba.it (A.N.); giuseppe.solarino@uniba.it (G.S.)

**Keywords:** anterior cruciate ligament, anterior cruciate ligament reconstruction, ACL, ACLR, infection, septic arthritis, graft failure, joint aspiration, arthroscopic debridement, arthrofibrotic

## Abstract

Infection is an uncommon side effect of arthroscopic surgery, and this percentage is higher in anterior cruciate ligament reconstruction (ACLR) surgery, where graft and fixation devices are used. Infections can not only lead to high re-admission rates and poor functional recovery of the knee but can also have a significant negative impact on the patient’s psychological and economic health, especially in athletes, as it can affect their sports career. It is important to be aware of the many risk factors, especially the manifestation of symptoms. These may sometimes be non-specific to the infectious pathology and common to other situations, such as the presence of a significant intra-articular hematoma. Septic arthritis after ACLR can occur at any time after surgery but typically presents acutely, while late manifestation is relatively rare. Diagnosis of infection is based on patient history, physical examination, laboratory parameters, and analysis of synovial fluid after joint aspiration, which is the gold standard for diagnosing post-operative infection. Once symptoms appear and the diagnosis seems certain, it is necessary to intervene quickly with arthroscopic debridement and long-term antibiotic treatment to try to save the graft and resolve the infectious situation to avoid graft failure and arthrofibrotic sequelae. The aim of this paper is to provide an overview of the epidemiology, pathogenesis, risk factors, clinical presentation, diagnostic evaluation, and current treatment guidelines of septic arthritis after ACLR surgery by analyzing recent literature, in particular meta-analyses and systematic reviews.

## 1. Introduction

Infection, with an estimated overall incidence of less than 1%, is an uncommon complication of arthroscopic surgery. This percentage is higher in ligament reconstruction surgery, where graft and fixation devices are used. In recent years, the number of reconstructive procedures and revisions following new injuries has increased significantly due to longer sporting careers and greater functional needs of patients. This increase is directly proportional to the incidence of infection. The condition is rare; it is important to obtain a diagnosis as soon as possible to prevent serious consequences such as joint damage due to the inflammatory response and the release of destructive enzymes from synoviocytes and leukocytes and consequences such as arthrofibrosis. Most infections following anterior cruciate ligament reconstruction (ACLR) are acute postoperative infections with a vague clinical presentation which may occur in uninfected patients with a large hematoma or in patients with inflammatory diseases. Symptoms may include fever, increased pain and stiffness, local erythema, swelling, warmth, fibrinous exudate, and other manifestations with variable severity [1].

Infections of the graft results in high hospital readmission rates and poor knee functional recovery. Consequently, the return to sports activities is also delayed and may have a significant negative impact on the patient’s psychology and income, especially for professional athletes. Knowing the risk factors, trying to avoid them, and taking appropriate action in the event of an infection can save the patient, the sports club, or the healthcare institution a considerable amount of money and also reduce the professional’s risk of being sued for negligence.

In the literature, most papers discuss arthroscopic infection in a generic way. Systematic reviews and meta-analyses focus on specific aspects of ACLR surgery without considering all pathological aspects of infection. The aim of this paper is to help readers understand ACLR infections by providing an overview of epidemiology, pathogenesis, risk factors, clinical presentation, diagnostic evaluation, and current treatment options guidelines and of septic arthritis after ACLR surgery. The impact of the infection and of the various treatments on the patient’s ability to return to their sporting activities is also evaluated.

## 2. Materials and Methods

The literature search was carried out on Pubmed, Medline, and Cochrane Library. Only studies published in the last 10 years from January 2013 were evaluated. The literature search was completed on 30 December 2023. All English-language studies were considered potentially eligible. The keywords used were as follows: “Anterior cruciate ligament infection”, “ACL infection”, and “knee arthroscopy infection”. A manual search of the reference lists of the selected publications was also performed to identify additional studies for potential inclusion. If full texts were not available, authors were contacted. Two reviewers (T.L. and I.R.) independently screened titles, abstracts, and full texts. Discrepancies between the reviews were thoroughly examined on the full-text screen to avoid inadvertent exclusion. Potential disagreements were resolved by a consensus decision mediated by a third reviewer (G.D.C.) to overcome potential discrepancies.

## 3. Results

Of the 3221 studies collected (400 from Medline, 2544 from Pubmed, 277 from Cochrane Library), 1652 were excluded, reviewing the titles and abstracts, and 1310 duplicates were removed using Rayyan (Johnson & Phillips, 2018, Newport, UK). The full text of the remaining 259 studies was examined in greater detail, and 57 studies were finally included in this narrative review. Studies involving other joints or surgical procedures on the knee other than ACLR were excluded; particular attention was paid to meta-analyses and systematic reviews focusing on specific areas of infectious pathology.

## 4. Epidemiology

Infection following ACLR remains a rare event, with rates ranging from 0.4% (in a community registry with 16,000 patients) [2] to 0.6% (in a recent systematic review) [3] and to 1.4% in a court study of over 1800 patients [4].

In a meta-analysis that looked only at ACL revision surgery, the MARS study group found that the overall rate of infection was 0.6%, and the use of allografts increased the odds of infection by 6.8 (95% CI: 0.9–54.5; *p* = 0.045) compared to autografts [5].

The pediatric and adolescent population has a modest (0.52%) infection rate following ACLR, which is comparable to that of young adults, and in particular, patients under the age of 15 appear to have a slightly lower infection rate (0.37%) [6].

## 5. Risk Factors

It is important to identify risk factors for surgical site infection (SSI) following ACLR as early as possible and to take preventive measures.

There is some evidence for specific risk factors: sex (males had significantly increased odds of SSI compared with females, *p* < 0.00001, OR 1.78, 95% CI 1.43 to 2.21), diabetes (*p* = 0.0003, OR 1.41, 95% CI 1.17 to 1.69), corticosteroid use history for immunological or neoplastic diseases (*p* < 0.00001, OR 4.80, 95% CI 2.61 to 8.84), previous knee surgery history (*p* = 0.0003, OR 2.86, 95% CI 1.62 to 5.05), professional athletes (*p* < 0.00001, OR 4.78, 95% CI 2.65 to 8.63), operating time (*p* = 0.005, 95% CI 2.49 to 13.75) surgery time was increased by approximately on average 8 min in those who experienced a SSI), revision surgery (*p* = 0.009, OR 1.63, 95% CI 1.13 to 2.37), and concomitant associated procedures such as lateral extra-articular tenodesis (*p* = 0.0001, OR 3.92, 95% CI 1.96 to 7.84). The reduced infection rate is considered to be an advantage of anatomic antero-lateral ligament reconstruction percutaneous procedure over iliotibial band procedures that require a larger lateral incision [7]. However, in high-volume centers where these procedures are performed more frequently, with less tissue damage and in a shorter time, the infection rate should be lower.

Two meta-analyses show different results about obesity and tobacco use: obesity with a BMI over 30 was associated with an 82% increase in the odds of SSI (*p* = 0.0005, OR 1.82, 95% CI 1.30 to 2.55) according to Zhao et al. and not significant correlation (*p* = 0.27, 95% CI, −1.58 to 0.44) according to Zhang et al. The first demonstrates statistical significance in the correlation between SSI and tobacco use (37% odds increase, *p* = 0.01, OR 1.37, 95% CI 1.08 to 1.75), unlike the second (*p* = 0.06, OR, 1.47, 95% CI, 0.99–2.18) [7,8]. There was no association between age, out- or inpatient surgery, concomitant meniscal suture, cartilage lesion, and the odds of SSI [7].

Patients with self-reported diabetes had an increased odds of infection by 28.6, *p* = 0.004, 95% CI: 5.5–149.9 [5]. A state of altered glycaemic control causes dysfunction of the immune system, particularly neutrophils and T lymphocytes, facilitates the inhibition of antioxidant mechanisms that constantly target free radicals and toxins and counteracts adaptative immunity, i.e., immunity mediated by antibodies.

Despite the heterogeneity of study data in the literature, professional athletes were significantly more likely to develop septic arthritis than patients who were not professional athletes (*p* < 0.00001). This statistically significant difference might be partially explained by the involvement of professional athletes in outdoor sports, causing the higher occurrence of cutaneous microscopic lesions. In effect, athletes who play sports involving frequent skin-to-skin contact, like rugby or football, have higher rates of skin infections, and turf burns and skin abrasions, which are frequently left untreated, offer a breeding ground for pathogenic bacteria [7,8,9]. Professional athletes are also subject to greater monitoring than amateur athletes, so infections in the latter group occasionally go undetected or unreported.

Different graft types also have different risks of infection: the risk is significantly lower using patellar bone-to-bone (BTB) autografts compared to hamstring autografts. There is no significant difference in the incidence of infections with autografts compared to allografts, which even show a 77% lower incidence of infections compared to the hamstring group (RR, 0.23; 95% CI, 0.097–0.54) [10].

These data are confirmed by another meta-analysis by Kenan Kuršumović et al. It was found that hamstring autografts were associated with a higher risk for infection rate (1.1%, CI = 0.9% to 1.2%) than BPTB autografts (0.7%, CI = 0.6% to 0.9%) [11].

Unlike in the past, as mentioned above, due to new techniques of irradiation and chemical processing, there is no significant difference in the incidence of infections after ACLR with autografts compared with allografts (RR, 1.035; 95% CI, 0.589–1.819) [10]; although some authors even claim that there was no difference in the likelihood of 90-day deep infection for processed versus unprocessed allografts (odds ratio = 1.36, 95% CI = 0.31–6.04) [12].

Contamination after harvesting and during graft preparation is the most accepted hypothesis due to several potential causal mechanisms, including the need for cannulated instruments for harvesting hamstring autografts, longer preparation times, the use of potentially bacterial multifilament sutures during graft preparation, and the placement of the graft harvest site directly over the tibial tunnel [13,14]. Offerhaus and colleagues, using microbiological culture and 16S rRNA-PCR, found that semitendinosus autografts were often contaminated with skin commensal bacteria at harvest [15]; furthermore, there is no evidence of contamination using different graft harvesting and preparation techniques [16].

Low infection rates (0.1%) have been reported in a few studies evaluating the quadriceps tendon [17], while porcine patellar tendon xenografts showed a higher rate of infection than human Achilles tendon allografts according to a randomized controlled clinical trial [18]. However, these studies were not systematic reviews or meta-analyses. Moreover, no studies evaluating infection rates in LARS synthetic grafts have been found [19].

A common clinical practice is to mark the skin around the incision with a marker pen. Often, the same marker is used to mark different areas of the graft during preparation.

Some authors have considered whether this practice could be a potential form of graft contamination. They sent the marker pen tip for culture after skin marking in 20 patients and found *Staphylococcus* species colonization in three cases. Therefore, it is recommended that the skin marker should be used only once and discarded after use, also considering its negligible cost [20].

Despite the World Health Organization (WHO) recommending antibiotic prophylaxis when a considerable amount of hardware is to be implanted [21], there is evidence supporting the use of prophylactic antibiotics in knee arthroscopic procedures to prevent postoperative infections (*p* = 0.05), but there are no studies available comparing ACLR with and without antibiotic prophylaxis. Most of them vary in the type of antibiotic prophylaxis used [22]. Since the graft is avascular in the first weeks after surgery [23], it behaves like a foreign body, and the screws or buttons are also small implants that can be colonized by pathogens. The most practical alternative is a single dose of intravenous antibiotic prophylaxis. Oral postoperative antibiotics should be avoided due to their adverse effects, limited absorption, and consequent lack of efficacy [24].

## 6. Microbiology

In ACLR infection, the most common isolated pathogens are *Staphylococcus* species (60–90%), with coagulase-negative staphylococci being the most common, followed by Staphylococcus Aureus [25]. Anaerobes such as *Peptostreptococcus* species and *Cutibacterium* (formerly *Propionibacterium*) species, Gram-negative bacteria (*Enterobacterales*, *Pseudomonas aeruginosa*), streptococci [26], and enterococci are other causal agents involved with a lower frequency. However, cases of rare bacteria have also been identified and described in case report studies. These include *Clostridium difficile*, which infected an immunocompetent patient undergoing ACLR with a hamstring autograft [27], *Granulicatella adiacens* [28], and *Nocardia* species (*Nocardia aobensis*, *Nocardia africana*, *Nocardia veterana*, and *Nocardia elegans*) [29].

Nontuberculous mycobacteria (NTM) infections are very rare and frequently have nonspecific symptoms and a slow course. Delays in diagnosis can result from the low index of suspicion; 6 cases identified were reported between 2010 and 2021 in database searching. The reported NTM species were *M. fortuitum*, *M. farcinogenes*, *M. abscessus* [30].

Fungal osteomyelitis after ACLR has also been described and is a serious complication because mucormycosis diagnosis is usually delayed; the first symptoms of infection appear after an average of more than 20 days (median time of 23 days after the ACLR), but often they do not appear until large bone destructive lesions are present. Contrary to what one might think, they did not occur in patients with weakened immune systems or in those with several major risk factors [31].

This may originate additional massive reconstructive surgeries with severe functional limitations for the patients. Identified fungal cultures are represented by Rhizopus species and Aspergillus species; there was bacterial overlap in some cases [31].

Culture-negative infections have been reported in up to 15–30% of cases following ACLR [25]. This can be caused by previous antibiotic therapy or by the presence of biofilm, a syntrophic community of microorganisms in which cells adhere to each other and often to a surface. Cells in the biofilm phenotype grow poorly on agar plates. A chemical or physical system should therefore be used to destroy them before analysis. However, this is not described in clinical practice to identify infections after ACLR.

## 7. Diagnosis

Septic arthritis following ACLR may appear at any time after surgery but typically presents either acutely while late manifestation is relatively infrequent.

Diagnosis of infection is based on patient history, physical examination, laboratory parameters, and analyses of synovial fluid following joint aspiration.

Suggestive systemic symptoms of infection are not specific, which may occur in uninfected patients with a large hematoma or in patients with inflammatory diseases. For example, Adult-Onset Still’s Disease (AOSD) can present with symptoms similar to those of septic arthritis. In fact, AOSD manifests itself with hyperthermia, often higher than 39° C, in most cases polyarthralgia, rash, lymphadenopathy, weakness, and sometimes splenomegaly and hepatomegaly.

The most important signs are prolonged knee discomfort, increased local temperature, tenderness to gentle percussion of the joint, recurrent or persistent knee swelling, and decreased range of motion (ROM). The following symptoms are indicative but not specific for infection: delayed recovery from physiotherapy, increased warmth or edema, and drainage from the incision site/portals (the tibial tunnel is most commonly involved). Purulent secretion, a sinus tract, or intraoperative detection of intraarticular pus are confirmative signs of infection. Low-virulence pathogens cause low-grade inflammation that manifests as chronic pain and sometimes arthrofibrosis [32].

The European Society of Sports Traumatology, Knee Surgery & Arthroscopy (ESSKA) and The European Bone and Joint Infection Society (EBJIS) have issued recommendations for the diagnosis and antimicrobial treatment of infections after ACLR [25,26,27,28,29,30,31,32,33].

Both C-reactive protein (CRP) and erythrocyte sedimentation rate (ESR) were helpful in determining the presence of a normal or septic joint. CRP was shown to be more sensitive and specific for the diagnosis of infection than ESR; it has high sensitivity in acute infections and low sensitivity in chronic infections. However, it should be interpreted with caution because resorption of hematoma can also cause a high or secondarily increasing CRP. CRP levels up to five times the normal limit are common in the month after an ACLR in males and in patients with chondral lesions treated with microfracture. A normal CRP does not exclude infection [34,35]. Serological analysis with white blood cell (WBC) count may be elevated but is not specific; in fact, the increase in total WBC count is indicative of inflammation and infection. However, it can be altered in several clinical conditions and could be normal or even decreased in some cases of sepsis. The relative WBC value can be helpful for evaluating which WBC population (Lymphocytes, monocytes sub-population, leukocytes) is mainly involved in the inflammatory process, allowing an etiological diagnosis [35].

Synovial fluid aspiration and cell count are the gold standard for diagnosing postoperative infection. A leukocyte count compatible with infection is considered a confirmatory criterion by many authors, but there is no consensus on cut-off values [36]. A normal leukocyte count (<2000/µL) rules out infection in most cases. A polymorphonuclear cell count >90% indicates a high probability of infection [36].

Synovial leukocyte count and Polymorphonuclear Cell Count are the most reliable tests for the diagnosis of septic arthritis, and a cut-off value of 28,100 cells/mL allows for a correct diagnosis in approximately 85% of cases, while a value of >40,000 cells/mL shows a specificity of 100% [37].

However, conventional culturing on agar plates is considered the main test for isolating the specific pathogen. Inoculation of synovial fluid into pediatric blood culture bottles showed a higher diagnostic yield in terms of pathogen detection compared to the traditional agar plate method [37].

Antibiotic pretreatment may interfere with the microbiological examination and should be avoided until synovial fluid is harvested if the patient is not septic.

In patients pretreated with antimicrobials before joint aspiration with negative culture results, bacterial DNA can be identified by polymerase chain reaction (PCR).

None of biomarkers, such as synovial fluid glucose, alpha-defensin, synovial C-reactive protein (sCRP), interleukin-6 (sIL-6), leukocyte esterase, and lactate dehydrogenase, has yet been validated in a representative study for infections after ACLR so the literature does not support their introduction in clinical practice [25,26,27,28,29,30,31,32,33,34,35,36,37].

## 8. Antibiotic Presoaking

There are numerous systematic reviews and meta-analyses that evaluate the treatment and antibiotic presoaking of the graft before implantation. Both Naendrup et al. conclude that evaluating eight studies with 5075 patients, vancomycin-soaking reduced the incidence of postoperative septic arthritis following primary ACLR. The common odds ratio was 0.04 [0.01–0.16], favoring that patients receive additional vancomycin-soaking of grafts [38].

The results are confirmed by Hu et al.: patients treated with vancomycin had significantly lower infection rates (0.09% versus 0.74%; OR 0.17) [39] by Xiao, who, in a meta-analysis of over 20,000 patients, found a lower infection rate in the vancomycin group (0.013% versus 0.77%; odds ratio 0.07) [40], and by Figueroa et al. who showed a 93% risk reduction compared with control patients [41].

All studies used a vancomycin concentration of 5 mg/mL, and the duration of graft-soaking varied between from 10 to 15 min.

In the past, many clinicians had concerns regarding the routine use of vancomycin in ACLR due to its toxic effect on chondroblasts, osteoblasts, and tenocytes and due to possible negative effect on the biomechanical properties of the graft [23,42], but the synovial concentrations of vancomycin rays as demonstrated by Pfeiffer and colleagues were not toxic (23.23 ± 21.68 µg/mL on average, while concentrations necessary for toxicity was >1000 µg/mL) [43]. Recent biomechanical studies showed that vancomycin presoaking does not affect the immediate properties of semitendinosus tendons [44] and is a highly cost-effective prophylactic measure for infection prevention [45].

Presoaking of hamstring autografts with gentamicin intraoperatively is a good alternative to vancomycin in the prevention of infection [46], although there are no comparative meta-analyses or systematic reviews.

## 9. Management

Early clinical suspicion is the most important factor in achieving a timely diagnosis. In recent septic arthritis, most of the literature favors graft retention, arthroscopic debridement, and long-term antibiotics as the treatment of choice, while if diagnosis and treatment are delayed, the risk of removing the graft and damaging the articular cartilage is greater [47]. Conservative treatment with antibiotics with or without joint irrigation was found to be less successful than arthroscopic debridement, with 30% to 43% of patients needing conversion to arthroscopic debridement, so it is not recommended based on the high failure rates. Surgical treatment is necessary to wash out proteolytic enzymes and toxins that cause chondrocyte degeneration and should be performed immediately when the infection is considered [48].

Arthroscopic debridement included removal of the devitalized, inflamed, or necrotic tissue and removal of fibrin layers on the graft and sometimes on cartilage and synovium, followed by extensive irrigation using at least over 15 L of fluid. Fixation implants may require removal or exchange, although it is not always possible [48]. Samples of the synovial tissue have to be collected and sent for cultivation to identify the causing pathogen [49]. A minimum period of antibiotic treatment of four weeks is recommended; afterward, intravenous antibiotics change to culture-sensitive oral antibiotics for at least 3 weeks or until normalization of clinical and laboratory parameters [50]. There are no established guidelines on whether or not to retain the graft, and this depends on the surgeon’s judgment, taking into account the arthroscopic visualization of the graft, together with clinical and laboratory findings. Kuršumović, in a meta-analysis, shows a success rate of about 85**%** in saving the graft where this was the primary intention. Graft preservation is important as it avoids further complex reconstructive procedures with their associated morbidity and potential complications. Serious infections require multiple arthroscopic debridements or in case of absence of deteriorating symptoms with unresolved laboratory markers [51], arthrotomy should only be performed in limited cases.

Graft salvage is achievable in acute presentations but less so in late presentations.

The long-term risks of retaining grafts with persistent infection include the risk of cartilage loss, arthrofibrosis, and chronic graft failure due to prolonged exposure to the infectious agent, which may have compromised the graft’s resistance to tensile loading and delayed the integration process. These complications are not universally reported across all studies on graft retention. Wang found that retention was more successful where septic arthritis was diagnosed within seven days post-ACLR compared with those diagnosed later. Infection may not have been completely eradicated in all cases of graft salvage, as low-grade infection may remain dormant without clinical symptoms. However, early removal of the graft is described in some studies, and patients who underwent early graft removal and staged reconstruction demonstrated restored knee stability, and the sub-acuity of presentation was not associated with laxity [52]. Graft and hardware removal should be performed in cases of bony involvement of the tibia or femur (Gächter classification of joint infections type IV) [53]. After removal, graft reimplantation within 3 to 6 weeks has furthermore shown promising results without waiting the recommended 6 to 9 months if the following criteria are fulfilled: no bone involvement (no osteomyelitis), favorable clinical evolution, decreasing CRP (normal value is not required), absence of difficult-to-treat infections caused by a microorganism that is resistant to biofilm-active antibiotics. At the time of repeat ACLR, tissue cultures from the synovial membrane and bone tunnels must be obtained during surgery. Histopathological analysis of bone fragments taken from the tunnels can help to exclude osteomyelitis [33,34,35,36,37,38,39,40,41,42,43,44,45,46,47,48,49].

## 10. Return to Activity

Once the infection has been eradicated, one of the most important goals is to achieve optimal clinical and functional outcomes to return to daily life activity or sport.

Thirty-one patients who had been successfully treated for infection were evaluated by Gille et al. at a mean follow-up of 6 years. They found a Tegner activity score of 4.5, which was significantly lower than the preoperative values (6.5), a Lysholm score of 63.9 (range 25–91), and a mean subjective IKDC score of 63 (range 31–92) [54]. In another retrospective case study with a minimum follow-up of two years, the Lysholm score and IKDC subjective score of the included patients reached a mean of 80.0 ± 15.1 and 78.2 ± 16.6 points, respectively, after 60.3 ± 39.9 months. After their last treatment for septic arthritis, all patients reported a median return to the sport of 8 months (6–16 months) at a mean follow-up of 99.3 ± 29.7 months. Twenty-one (55.3%) of the patients had stopped. Out of the patients, eighteen (51.4%) reported a decrease in frequency compared to their pre-injury condition. At follow-up, eleven patients (26.3%) reported returning to their pre-injury sport at the same frequency. Return to sports rates were unaffected by graft management (first graft salvage vs. first graft removal with subsequent ACLR revision) [55]. In two case–control studies, Abdel-Azis et al. [56] and Bostrom Windhamre et al. [57] reported that 56% and 62.5% of their patients had returned to recreational activities without specifying the kind of sport.

At follow-up, the only patient-dependent factor that was statistically linked to a lower return to sports (RTS) rate was persistent ACL deficit.

## 11. Conclusions

Infection after ACLR remains a rare event, but it is important to be aware of the many risk factors. There is some evidence for specific risk factors, such as gender, diabetes, history of steroid use, history of previous knee surgery, sports, graft type, operation time, and concomitant procedures. The most commonly isolated pathogens are staphylococci, but attention must also be paid to rarer pathogens with more or less virulence and especially to fungal infections. Pre-soaking the graft with vancomycin significantly reduces the rate of infection, but the most important thing is to recognize early infection, which often presents with non-specific signs and symptoms. Culture and leukocyte cell counts from synovial fluid samples are the gold standard for diagnosis. Early intervention means arthroscopic debridement combined with prolonged targeted antibiotic therapy in order to preserve the graft, prevent potential sequelae such as osteomyelitis and arthrofibrosis, and ensure optimal outcomes after resolution of the infection. Further studies to validate new potential biomarkers and rapid tests would be useful. In addition, a new classification that takes into account not only anatomopathological arthroscopic findings but also systemic and synovial proteomic parameters could help in the decision-making process and therefore in describing a correct flow chart to follow in patients with infections after anterior cruciate ligament reconstruction.

## Data Availability

Not applicable.

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
