# Peer review of "Infection after Anterior Cruciate Ligament Reconstruction: A Narrative Review of the Literature"

_healthcare, 2024, doi:10.3390/healthcare12090894_

Round 1
Reviewer 1 Report
Comments and Suggestions for Authors
The Authors provide a comprehensive review of the studies regarding infection after ACL reconstruction. The following comments are provided to improve the clarity of the manuscript and broaden the number of the potential readers. Due to the large amount of comments and suggesions for change, it is possible that not all issues have been addressed in this review.
Line 40: Suggest "a range of severity" instead of "varying severity.
Line 41-43: Suggested wording: Infection of the graft results in high hospital readmission rates and poor knee functional recovery. Consequently, the return to sport activities is also delayed and may have a significant, negative impact on the patient’s psychology and income, especially for professional athletes.
Line 45: Suggests "in a generic way. Systematic reviews
Line 47: Suggests "all pathological aspects of infection".
Line 49: Suggests "The impact of the infection and of the various treatments on the patients' ability to return to sport activities is also evaluated".
Line 64: Suggests "Potential disagreements were resolved by consensus decision mediated by a third reviewer (G.D.C.)".
Line 88: Please specify "steroid use": therapeutic corticosteroids for immunological or neoplastic diseases, anabolic steroids or else.
Line 91: If SSI is not used before, it needs full wording.
Line 110: Suggests "adaptive immunity" instead of "humoral immunity"
Lines 114-116: Suggested change "This statistically significant difference might be partially explained by the involvement of professional athletes in outdoor sports causing higher occurrence of cutaneous microscopic lesions".
It is suggested that the Authors provide some explanation of the temporal sequence of events between the skin lesion(s) and the infection of the ACL graft.
Lines 119-121: in the latter group instead of the former group? it looks that the unreported cases of infection should occur in the non-pro athletes group and not the opposite.
Line 127: Suggests "et al. It was found"
Line 130: Suggests "as mentioned above"
Lines 148-149: Suggested change: However, these studies did not include a meta-analysis. Moreover, no studies evaluating infection rates in LARS synthetic grafts have been found [19].
Comment: the Authors did not find the studies or were not found in ref 19? Please clarify.
Lines 156-157: Suggested change: "should be used only once and discarded after use, also..."
Line 161: Suggested change: "but there are no studies available comparing ACLR with and without antibiotic prophylaxis.
Lines 162-165: It is unclear what the Authors mean by "Since the graft is avascular, it behaves like a foreign body and the screws or buttons are also small implants. [Comment: the graft is subject to variable cell repopulation]. The most practical alternative is a single dose of antibiotic prophylaxis [Comment: meaning of single dose unclear]. Oral postoperative antibiotics should be avoided due to their adverse effects, limited absorption, and consequent lack of efficacy [23].
Lines 169-171. Suggested change: "In ACLR infection, the most common isolated pathogen is the Staphylococcus species (60–90%), with coagulase-negative staphylococci being the most common followed by Staphylococcus aureus [24]."
Line 172: Suggests "gram negative bacteria"
Line 174: The topic is ACLR and not septic arthritis.
Line 182: M. fortuitum, M. farcinogenes, M. abscessus is repeated twice.
Line 184: Fungal osteomyelitis after ACLR has also been described and is a serious complication. [Comment: At the end of the paragraph the relevant reference is missing]
Lines 190-198: Comment: These paragraphs need to be rephrased. It is unclear the possible role of a biofilm and of its cell culture. Reference(s) is missing.
Line 201: Comment: All ALCR infections cause septic arthritis?
Line 217: Please describe the difference between low and high virulence and high- and low-grade inflammation.
Lines 227: Please explain how the level of CRP can be associated to "less experienced surgeons".
Line 297: "Cultures [Samples instead of cultures?]of the synovial tissue have to be collected and sent for cultivation to identify the causing pathogen"
Line 329: "Histopathology of the tunnels may be of help in ruling out osteomyelitis". Please be more specific regarding on how histopathological examination rule out osteomyelitis and whch specimens would be needed.
Line 350: "lower RTS rate was persistent ACL deficit. RTS should have non-abbreviated form if it is the first citation.
Line 365: In addition, a new classification that takes into account not only anatomopathological arthroscopic findings [Which ones?]
Comments on the Quality of English Language
There are many imprecisions and need to rewarding for numerous paragraphs, some of them addressed in this review.
Reviewer 2 Report
Comments and Suggestions for Authors
Firstly, I would like to thank you for giving me the opportunity to review this interesting paper.General comments can be found here and more specific comments on the attached file.
Even if this is an interesting and flexible study, presented as a narrative review on infection after anterior cruciate ligament reconstruction some sessions need to be generously revised. English used are readable and but minor editing is required. The points, which need to be enhanced include:
Abstract: a comprehensive presentation of the interesting aspects of the study, the aim should be more clearly presented.
Introduction: The extension of this session is limited. Some interesting topics could be enhanced with additional literature according to the comments. Some paragraphs from the discussion could be incorporated in this one.
Methodology: Some important points of this session could be enhanced. Exclusion and inclusion criteria should be added in order the study to be further clarified. Some details related with study design should be added.
Results: This session could be enhanced. A more extended presentation or a flow diagram contribute to a more readable narrative review and clarify the criteria used for the studies selected. If a duplicate program has been used, this should be declared.
Discussion: An extended presentation of the literature regarding the points under investigation. Some paragraphs of this session must be better organized in order not to be confused and they could be incorporated in the introduction session. Paragraphs analyzing known literature could be shortened and these used to explain the results of this study should be presented with tables, summarizing results. References could be added in order to support the literature used.
Conclusions: Conclusions should be focused on what the results of this study can offer to the clinicians and whether main goals of this study have been achieved.

minor editing is needed
Reviewer 3 Report
Comments and Suggestions for Authors
This is an interesting narrative review on knee septic arthritis following arthroscopic anterior cruciate ligament reconstruction (ACLR). The authors provided a comprehensive outlook on the topic. However, some edits are needed before considering publication. Please find my comments below.
- According to the title and the study design, the authors have performed a narrative review, although they report (lines 53-54) to have followed PRISMA guidelines. However, the PRISMA statement includes several different items that have not been apparently considered. First, the author should clarify if they intend to perform a narrative review (with PRISMA guidelines not applicable), a scoping review (i.e., a narrative review with a systematic approach, which seems to better fit this manuscript, https://prisma-statement.org/Extensions/ScopingReviews), or a systematic review. In the last case, the paper should be completely reorganized.
- Line 29: Rather than a "side effect", I would define infection as a "complication" of arthroscopic surgery.
- Line 30: What kind of reconstructive surgery are you referring to? Cartilage repair, tendon repair, ligament repair, meniscus repair, anterior cruciate ligament reconstruction? These encompass different implantable devices and techniques. Please specify.
- Line 44: Please provide a reference in support of the statements within this paragraph.
- Line 47: Please provide the references of the papers you are referring to in this line.
- Lines 55-56: Why were only papers published in the last decade included? This seriously limits the assessment of the available evidence.
- Line 75: Please express this number in digits.
- Lines 87-93: It would be useful to report odds ratios and 95% confidence interval to get an estimate of the magnitude of reported risks. Same in lines 100-104.
- Line 99: Please provide a reference in support of this statement.
- Lines 108-111: Please provide a reference in support of this statement.
- Lines 119-121: May this also depend on the fact that ACLR is more often performed in the athletic population, which thus constitutes a larger portion of ACLR patients?
- Lines 84-166: It would be useful to summarize the risk factors described here in a Table structured in three different columns (risk factor, magnitude, prevention strategy).
- Line 200: What about imaging findings of patients affected by septic arthritis post-ACLR? May the authors report on that?
- Line 219: Please spell out "ESSKA" and "EBJ".
- Line 234: Please provide references in support of the statements within this paragraph.
- Line 239: Please provide a reference in support of this statement.
- Line 247: Please provide references in support of the statements within this paragraph.
- Lines 248-255: Please provide references in support of the statements within these lines.
- Line 293: Please provide references in support of the statements within this paragraph.
- Line 315: Please provide references in support of the statements within this paragraph.
Comments on the Quality of English LanguageThe text should be inspected for minor but frequent syntax and grammar errors.
Round 2
Reviewer 2 Report
Comments and Suggestions for Authors
I would like to thank you for giving me the opportunity to review this interesting paper, which has been revised according to the comments.
Even if, this is a flexible study, the revision has enhanced its content and most of its sessions, making it even more readable and interesting. The aims and methodology of the study have been emphasized and introduction session has been extended, as suggested. Exclusion criteria have been added, while discussion session remains well-organized following revision. Finally, English and reference sessions have been improved, too.
Author Response
Thank you so much, your suggestions have been very helpful
Reviewer 3 Report
Comments and Suggestions for Authors
The authors have significantly improved the quality of their manuscript according to the reviewers' comments. Please find attached some additional minor edits that should be implemented before considering publication.
- Since the design of this review is narrative, there is no need to state the explicit methodology since this is more suitable for scoping and systematic reviews and would be more appropriate to expand on search strategies, data extraction, implementation of a PRISMA flow chart, and so on. But considering that this is not the aim of this paper, I suggest removing lines 58 to 78, removing the "Discussion" header and changing to headers all the subheadings now in italics.
- Please add the Table as suggested; this would increase your manuscript's readability and summarize the main points of your review.
